# Fitness Costs and Incomplete Resistance Associated with Delayed Evolution of Practical Resistance to Bt Crops

**DOI:** 10.3390/insects14030214

**Published:** 2023-02-21

**Authors:** Yves Carrière, Bruce E. Tabashnik

**Affiliations:** Department of Entomology, University of Arizona, Tucson, AZ 85721, USA

**Keywords:** *Bacillus thuringiensis*, sustainability, Bt soybean, Bt corn and Bt cotton, genetically engineered, genetically modified, pest adaptation, resistance management, resistance syndrome, transgenic crops

## Abstract

**Simple Summary:**

Crops genetically engineered to produce insect-killing proteins from the soil bacterium *Bacillus thuringinsis* (Bt) have been used since 1996 to improve pest control. Intensive use of Bt crops has resulted in the evolution of practical resistance in some pests, which reduces the efficacy of Bt crops and has detrimental practical consequences for pest management. To better understand factors underlying this evolutionary response, we analyzed data from the literature to evaluate the association with the evolution of practical resistance for two pest traits: fitness costs and incomplete resistance. Fitness costs are negative effects of resistance alleles on pest fitness in the absence of Bt toxins. Incomplete resistance is the reduced fitness of resistant insects on Bt plants relative to non-Bt plants. Our results show that lower fitness costs and the higher survival of resistant pests on Bt plants relative to non-Bt plants are associated with the evolution of practical resistance. Together with previous studies showing that nonrecessive inheritance is associated with practical resistance, the results here identify a syndrome associated with the evolution of practical resistance to Bt crops. Further research to characterize this resistance syndrome and determine its evolutionary consequences could be useful to sustain the efficacy of Bt crops.

**Abstract:**

Insect pests are increasingly evolving practical resistance to insecticidal transgenic crops that produce *Bacillus thuringiensis* (Bt) proteins. Here, we analyzed data from the literature to evaluate the association between practical resistance to Bt crops and two pest traits: fitness costs and incomplete resistance. Fitness costs are negative effects of resistance alleles on fitness in the absence of Bt toxins. Incomplete resistance entails a lower fitness of resistant individuals on a Bt crop relative to a comparable non-Bt crop. In 66 studies evaluating strains of nine pest species from six countries, costs in resistant strains were lower in cases with practical resistance (14%) than without practical resistance (30%). Costs in F_1_ progeny from crosses between resistant and susceptible strains did not differ between cases with and without practical resistance. In 24 studies examining seven pest species from four countries, survival on the Bt crop relative to its non-Bt crop counterpart was higher in cases with practical resistance (0.76) than without practical resistance (0.43). Together with previous findings showing that the nonrecessive inheritance of resistance is associated with practical resistance, these results identify a syndrome associated with practical resistance to Bt crops. Further research on this resistance syndrome could help sustain the efficacy of Bt crops.

## 1. Introduction

Transgenic crops that produce toxins from the soil bacterium *Bacillus thuringiensis* (Bt) can reduce insect feeding damage on crops including corn, cotton, cowpea, eggplant, soybean, and sugarcane planted on large and small farms [1,2,3,4,5]. Advantages of Bt crops include a lower reliance on insecticide sprays for pest control [6,7,8,9], reduction in pest pressure across agricultural landscapes [8,10,11,12], and pest eradication [13]. Use of Bt crops has regularly expanded worldwide since their introduction in a few countries in 1996 [14]. Accordingly, many countries have developed governance systems to preserve the benefits provided by Bt crops [15]. Nevertheless, the evolution of resistance to Bt crops by pests has accelerated worldwide [14,16]. Practical resistance is defined as field-evolved resistance that reduces the efficacy of Bt crops and has practical consequences for pest control [17]. Thus far, at least 26 cases of practical resistance involving 11 pest species and nine Cry toxins have occurred in seven countries [14]. 

The refuge strategy is central for delaying the evolution of resistance to Bt crops. Refuges are areas of cultivated or non-cultivated non-Bt host plants that produce susceptible insects that can mate with resistant individuals surviving on Bt crops. Under ideal conditions, resistance to single-toxin crops is recessively inherited [18], or redundant killing is effective for multi-toxin Bt crops called pyramids [19,20]. Redundant killing is defined as the killing of individuals that are resistant to one toxin produced in a pyramid by another toxin in the pyramid [21,22]. When these conditions are met and the frequency of resistance alleles is low, most resistant individuals surviving on Bt crops mate with susceptible individuals from refuges and the resulting heterozygotes cannot survive on Bt crops. This reduces the heritability of resistance (i.e., the resemblance between resistant parents and their offspring) and delays the evolution of resistance [23,24].

In a review of resistance monitoring data for single-toxin Bt crops, resistance was recessively inherited in 67% of cases (6 of 9) where the percentage of resistant individuals remained below 1% but was nonrecessive in five cases where practical resistance occurred [25]. In another study primarily involving single-toxin crops, resistance was more recessive in pests without field-evolved resistance than in pests with field-evolved resistance [26]. These findings support the theoretical prediction that refuges coupled with high-dose crops can delay the evolution of practical resistance. 

The difference in average fitness among resistance genotypes on Bt crops and refuge plants governs the evolution of resistance. Factors influencing the fitness of resistance genotypes include the abundance of refuges, dominance of resistance, redundant killing, magnitude and dominance of fitness costs, and incomplete resistance [19,20,24,27,28,29]. Fitness costs (hereafter costs) are defined as trade-offs in which alleles conferring resistance to a Bt toxin have deleterious effects on fitness components in the absence of Bt toxins [24,28]. Incomplete resistance is defined as resistance in which resistant insects have lower fitness on a Bt crop than on a comparable non-Bt crop [24,27,28,30]. Mathematical models show that in theory, costs and incomplete resistance can jointly delay or reverse the evolution of resistance to Bt crops [27,30,31,32,33,34]. However, it has been unclear if these factors have actually influenced the evolution of practical resistance. Here, we review and analyze data on costs and incomplete resistance in cases where pests have or have not evolved practical resistance to single-toxin crops. The results indicate that higher costs and lower survival on Bt crops relative to non-Bt crops are associated with delayed evolution of practical resistance to Bt crops. 

## 2. Materials and Methods

### 2.1. Literature Review

The literature reviews conducted for this paper were completed by 21 October 2022. Details for statistical analysis used to compare costs and incomplete resistance between cases with and without practical resistance are provided in Appendix A.

### 2.2. Fitness Costs

We used the Scopus database to review the literature available on costs for the 24 pest species for which responses in the field to Bt crops were previously categorized as practical resistance, early warning of resistance, or no decrease in susceptibility [14]. We also reviewed papers on costs associated with Bt resistance cited by Gassmann et al. [28] and Huang [26]. Our review identified 66 studies that report costs for strains of nine pest species that were selected in the laboratory for resistance to one of ten Bt toxins produced by Bt crops (nine Cry toxins and Vip3Aa, Appendix A). From these 66 studies, we produced a dataset consisting of 98 cases from six countries, where each case includes the cost data for a Bt-resistant strain of the pest in a particular country and whether or not that pest evolved practical resistance to at least one Bt crop in that country according to Tabashnik et al. [14]. For a subset of 68 cases, cost data were also available for the F_1_ progeny obtained by crossing a resistant and susceptible strain (Appendix A).

We did not consider costs of resistance to more than one Bt toxin or costs evaluated with selection experiments [28], because too few studies of pest species monitored for resistance [14] were available to perform meaningful analyses. For studies reporting fitness components in figures, we used WebPlotDigitiser (https://automeris.io/WebPlotDigitizer/#:~:text=WebPlotDigitizer%20is%20a%20semi-automated,-platform%20(web%20and%20desktop (accessed on 12 February 2023)) to extract data on fitness components of the susceptible and resistant strains, as well as the F_1_ progeny when available. Because costs were previously found to be lower on an artificial diet than plants [28], we evaluated costs in strains fed an artificial diet, corn, or cotton. Each cost estimate considered in analyses involved one strain of one pest species from one country that was resistant to one Bt toxin and was fed one of the three diets. 

We used a fitness component method [28] to analyze costs. For components positively associated with fitness (e.g., survival, pupal weight, and fecundity), we calculated the fitness ratio W_R/S_ as W_R_/W_S_, where W_R_ and W_S_ are the mean values of the fitness component for the resistant and susceptible strains, respectively. For development time, the only component negatively associated with fitness, W_R/S_ was 1 − [(W_R_ − W_S_)/W_S_] when the development time was greater for the resistant strain than the susceptible strain, and 1 + [(W_S_ − W_R_)/W_S_] when the development time was smaller for the resistant strain than the susceptible strain. When relevant data were available, we calculated the fitness ratio W_RS/S_ as W_RS_/W_S_, by replacing W_R_ in the equations above with W_RS_, which is the mean value of the fitness component for the F_1_ progeny from matings between resistant and susceptible strains. For each of the fitness components, cost is 100% multiplied by one minus the fitness ratio. For example, if W_R/S_ = 0.80, the fitness cost is 20%.

Two types of data were used to estimate costs: R_o_ (i.e., the reproductive rate, the number of offspring produced per female and generation) and one or more individual fitness components. When R_o_ was reported, we used it to estimate costs. Values of R_o_ were used to calculate W_R/S_ or W_RS/S_ when a significant difference in R_o_ between the resistant strain or F_1_ progeny and the susceptible strain was reported, e.g., [35], or at least one of the fitness components used to calculate R_o_ differed significantly between the resistant strain or F_1_ progeny and the susceptible strain, e.g., [36,37]. When neither of these criteria were met, e.g., [38], W_R/S_ or W_RS/S_ was set to 1. When R_o_ was not reported, we used all fitness components that differed significantly between the resistant strain or F_1_ progeny and the susceptible strain to calculate W_R/S_ or W_RS/S_. In such a case, W_R/S_ or W_RS/S_ was the product of the fitness ratio for the relevant fitness components. For example, if the fitness ratio for survival, pupal weight, and development time is, respectively, 0.90, 0.95, and 0.80 for the resistant strain, then W_R/S_ is 0.90 × 0.95 × 0.80 = 0.68, indicating an overall cost of 32%. W_R/S_ or W_RS/S_ was set to 1 when there was no significant difference in any of the measured fitness components between the resistant strain or F_1_ progeny and the susceptible strain. Thus, values of W_R/S_ or W_RS/S_ < 1 indicate a significant cost, values = 1 no cost, and values > 1 a significant fitness advantage for the resistant strain or F_1_ progeny over the susceptible strain.

Most papers that reported R_o_ also reported r_m_, the rate of increase, which is calculated as r_m_ = ln R_o_/T, where T is the mean generation time, e.g., [36,37]. We did not use r_m_ to calculate W_R/S_ or W_RS/S_, because r_m_ underestimates costs. Fitness costs consider differences in fitness between the resistant strain or F_1_ progeny and the susceptible strain during a single generation [28] because such parameters are related to the selection differential required to predict resistance evolution, e.g., [27]. Costs are underestimated when based on r_m_ because r_m_ considers the demographic impact of life history traits over several generations of exponential growth. For example, W_R/S_ = 0.80 if R_o_ = 100 for a susceptible strain and 80 for a resistant strain, which corresponds to a 20% fitness cost. For simplicity, assume that T is 10 for the susceptible and resistant strain (so T does not contribute to the difference in r_m_ between the strains). This, respectively, yields r_m_ = 0.46 and 0.44 for the susceptible and resistant strain and W_R/S_ = 0.96, which corresponds to a 4% fitness cost. Use of r_m_ instead of R_o_ in this example underestimates costs by 5-fold. 

Some studies estimated costs in the resistant strain or F_1_ progeny under different conditions. For example, costs in the same resistant strain were evaluated at several temperatures [39]; in males and females [40]; in several experiments conducted in the laboratory, greenhouse, or field [41]; for different corn hybrids [42] or plant phenology [43]; or under diapausing or non-diapausing conditions [44]. In such cases, cost estimates for a strain under the different conditions were averaged and the mean value of W_R/S_ or W_RS/S_ was used in analyses. The resistant and susceptible strains used in fitness components studies could differ for reasons not related to resistance. Two major sources of variation are differences in geographic origin linked to divergent selection on life history traits, and the variation in inbreeding depression associated with laboratory rearing conditions and history. Inbreeding depression is defined as the decline in value of one or more fitness components in offspring of related individuals [45] Strains with a similar genetic background should be least affected by factors unrelated to resistance. Thus, comparisons between related strains are expected to provide better estimates of the occurrence, magnitude, and dominance of costs than comparisons between unrelated strains [28].

The genetic background of a resistant strain that is backcrossed with a susceptible strain is expected to be 75, 88, 94, and 97% similar to the susceptible strain with 2, 3, 4, or 5 backcrosses, respectively. We considered the susceptible and resistant strain to be related when the resistant strain was selected for resistance from the susceptible strain or when ≥4 backcrosses were used to make the genetic background of the resistant strain similar to the susceptible strains. For the investigation of costs of resistance to Cry1Ac in *Pectinophora gossypiella*, crosses between resistant and susceptible strains generated heterogeneous strains for which PCR was used to identify resistant and susceptible genotypes segregating in the same genetic background, e.g., [46]. Accordingly, these cases were considered under the category of related strains.

### 2.3. Incomplete Resistance 

We used a subset of the 66 papers reviewed for costs (as described above) to review data on incomplete resistance (IR). Our review of incomplete resistance included 24 studies and 29 cases involving resistance to one of eight Bt toxins (seven Cry toxins and Vip3Aa) in seven pest species from four countries (Appendix A). We measured IR as survival of a resistant strain on a Bt crop divided by its survival on a corresponding non-Bt crop. We calculated IR for Bt-resistant strains fed Bt and non-Bt corn or Bt and non-Bt cotton. Thus, a value of IR = 1 indicates that survival was the same on the Bt and non-Bt crop (i.e., complete resistance), whereas values less than 1 indicate that survival was lower on the Bt crop than the non-Bt crop (i.e., incomplete resistance). As for costs, IR estimates obtained under different conditions for the same strain were averaged and the mean value of IR was used in analyses. IR estimates considered in analyses involved one strain of one pest species from one country that was resistant to one Bt toxin and fed on Bt and non-Bt corn or cotton.

### 2.4. Computer Simulations of Evolution of Resistance

Our analyses of previously published data show that fitness costs are lower and survival on Bt crops relative to non-Bt crops is higher in cases with practical resistance than cases without practical resistance (see Results). Previous studies show that the dominance of resistance is higher in cases with practical resistance than in cases without [25,26]. Here, we used a deterministic simulation model previously developed for two-toxin pyramided crops [43,47] to assess how these differences may affect the evolution of resistance to single-toxin crops. We set the frequency of the allele conferring susceptibility at one of the resistance loci to 1, which converted the two-toxin model into a model with one locus at which two alleles confer either resistance (*r*) or susceptibility (*s*) to the Bt crop. This is the simplest assumption about the genetic basis of resistance and a reasonable starting point because resistance to a single toxin produced by the Bt crop is often controlled primarily by alleles at one locus [48,49,50,51]. The dominance of resistance was measured with the parameter *h*, for which 0 indicates recessive and 1 indicates dominant inheritance [52]. We set resistance as recessive (*h* = 0) for cases without practical resistance and nonrecessive (*h* = 0.26) for cases with practical resistance. We chose *h* = 0.26 to reflect the dominance of survival from neonate to adult on Cry1Ac cotton in *Helicoverpa zea* [47]. This *h* value is within the range of dominance estimates (0.1–1.6) for other cases with practical resistance to single-toxin Bt crops [25,26]. We simulated recessive costs based on the means calculated here for observed cases with and without practical resistance (14 and 30%, respectively) as well as a hypothetical scenario with no cost. For each cost value, we simulated IR based on the means calculated here for observed cases with and without practical resistance (0.76 and 0.43, respectively) as well as a hypothetical scenario with IR = 1 (complete resistance). The dominance of resistance was kept constant in simulations of cases with *h* = 0.26 by adjusting the fitness of *rs* across values of IR (Appendix A). The initial frequency of the *r* allele was 0.001 [14,24]. The time to resistance was the number of generations for the *r* allele frequency to exceed 0.5. Simulations were run until this threshold was exceeded or for 400 generations.

## 3. Results

### 3.1. Comparison of Costs between Cases with and without Practical Resistance

#### 3.1.1. Costs in Resistant Strains

Analysis of the data for fitness of resistant strains relative to susceptible strains in the absence of Bt toxins (W_R/S_) indicates costs were lower in cases with practical resistance than in cases without practical resistance. The basic statistical model used to compare W_R/S_ between cases with and without practical resistance included strain relatedness (related or not), food type (artificial diet, corn, or cotton), and species and country (e.g., *Helicoverpa armigera* from Australia) nested within food type as explanatory variables (see Appendix A for details and other non-significant effects tested). Analysis of this model revealed one outlier (studentized residual = 3.93). This observation was for *H. armigera* from Australia resistant to Cry2Ab [53]. W_R/S_ was 2.46 for this case, indicating that fitness was 2.46-fold higher for the resistant strain than the susceptible strain. Although this extreme observation did not have qualitative effects on the significance of factors included in the model, it was excluded from analyses because it did affect cost estimates for several explanatory variables. 

W_R/S_ did not differ significantly between related and unrelated strains (df = 1, 76, *F* = 1.33, *P* = 0.25). The average cost estimate was 12% for related strains (back-transformed least squares mean W_R/S_ = 0.88, 95% confidence interval = 0.75–1.00) and 21% for unrelated strains (W_R/S_ = 0.79, 0.68–0.98). Food type was not significantly associated with W_R/S_ (df = 2, 76, *F* = 1.30, *P* = 0.28), after taking into account effects of strain relatedness. Cost estimates on an artificial diet, corn, and cotton were, respectively, 23% (W_R/S_ = 0.77, 0.64–0.89), 18% (W_R/S_ = 0.82, 0.67–0.96), and 8% (W_R/S_ = 0.92, 0.78–1.06). Contrasts between the average W_R/S_ on the diet and each host plant did not provide evidence for lower costs on corn (df = 1, 76, *F* = 2.57, *P* = 0.11) or cotton (df = 1, 76, *F* = 0.29, *P* = 0.59). 

After controlling for effects of strain relatedness and food type, the variation among species and country nested within food type (Table 1) was not significant (df = 17, 76, *F* = 1.23, *P* = 0.26). 

Across diets, the average cost for cases without practical resistance was 21% (W_R/S_ = 0.79, 0.66–0.92), which was not significantly higher than the average cost of 14% for cases without practical resistance (W_R/S_ = 0.86, 0.74–0.98; one-tailed contrast of W_R/S_ estimates, df = 1, 76, *F* = 0.65, *P* = 0.42). However, the average cost for *Diatraea saccharalis* on the diet and corn was 0%, which is low relative to other cases without practical resistance (Table 1, range 20–47%). A contrast excluding estimates of W_R/S_ for *D. saccharalis* indicates that costs were higher in cases without practical resistance (cost = 30%, W_R/S_ = 0.70, 0.58–0.81) than in cases with practical resistance (cost = 14%, W_R/S_ = 0.86, 0.76–0.96; one-tailed contrast, df = 1, 76, *F* = 2.64, *P* = 0.054). Contrasts not excluding the extreme W_R/S_ value for *H. armigera* (see above) provided qualitatively similar conclusions (log-transformed W_R/S_, *P* = 0.21 and *P* = 0.062 for contrasts including or not including data for *D. saccharalis*, respectively). Below, we show that costs were probably underestimated in *D. saccharalis* for methodological reasons.

#### 3.1.2. Costs in F_1_ Progeny

Analysis of the basic statistical model (see Appendix A for details and other non-significant effects tested) indicates that costs in the F_1_ progeny relative to the susceptible strains were recessive and did not differ between cases with and without practical resistance. To improve the estimate of cost in the F_1_ progeny, we excluded four outliers that had marginally significant effects in analyses. These cases had studentized residuals of 3.00, 2.94, −3.25, and −3.10, respectively, for *H. armigera* from Australia resistant to Cry2Ab [53] (W_RS/S_ = 1.51), *D. saccharalis* from the US resistant to Cry1Ab [54] (W_RS/S_ = 1.85), and the Pr and FL strains of *Spodoptera frugiperda* from the US resistant to Cry1F [55] (W_RS/S_ = 0.52 and 0.64, respectively).

Mean values of W_RS/S_ did not differ significantly from 1 for related strains (1.05, 95% confidence interval 0.99–1.11) or unrelated strains (0.97, 0.92–1.02), indicating that no cost was detected for F_1_ progeny relative to susceptible strains. In addition, W_RS/S_ did not differ significantly between related and unrelated strains (df = 1, 47, *F* = 2.90, *P* = 0.095). After correcting for strain relatedness, food type was not significantly associated with costs (df = 2, 47, *F* = 1.51, *P* = 0.23). For each of the three food types, W_RS/S_ did not differ significantly from 1 (i.e., no cost detected). The average W_RS/S_ was 0.97 (0.91–1.02), 1.02 (0.97–1.08), and 1.04 (0.97–1.03) on cotton, artificial diet, and corn, respectively. After controlling for effects of strain relatedness and food type, the variation among species and country nested within food type was not significant (Table 1) (df = 13, 47, *F* = 1.23, *P* = 0.29). Contrasts with (*P* = 0.41) or without (*P* = 0.15) W_RS/S_ estimates for *D. saccharalis* did not provide evidence that costs differed between cases with or without practical resistance. The mean W_RS/S_ excluding data for *D. saccharalis* (see evidence below that costs in F_1_ progeny were overestimated in *D. saccharalis*) was 1.03 (0.97–1.08) and 0.96 (0.89–1.03) for cases with and without practical resistance, respectively. The 95% confidence intervals for these estimates overlap 1, indicating that, on average, costs were recessive in the F_1_ progeny. Contrasts for analyses including the four outlier values of W_RS/S_ (see above) provided qualitatively similar conclusions (*P* = 0.57 and *P* = 0.81 for contrasts with and without data for *D. saccharalis*, respectively).

#### 3.1.3. Frequency and Dominance of Costs in Unrelated vs. Related Strains

We analyzed data on the frequency of costs to further evaluate how cost estimates were affected by the use of unrelated or related strains. Analyses show that the occurrence of costs in resistant strains was similar when related or unrelated strains were used. Costs were detected (i.e., W_R/S_ < 1) in 19 cases (41%) and not detected (i.e., W_R/S_ ≥ 1) in 27 cases (59%) for related strains. Costs were detected in 29 cases (56%) and not detected in 23 cases (44%) for unrelated strains. The percentage of cases with costs detected did not differ significantly between related and unrelated strains (Fisher’s exact test, *P* = 0.16). 

Both W_R/S_ and W_RS/S_ were estimated for 68 cases, which allowed analysis of recessive and nonrecessive costs (Table 2). Costs affecting the F_1_ progeny were detected (i.e., W_RS/S_ < 1) in 5 cases (17%) and not detected (i.e., W_RS/S_ ≥ 1) in 25 cases (83%) for related strains. Costs affecting the F_1_ progeny were detected in 11 cases (29%) and not detected in 27 cases (71%) for unrelated strains. The percentage of cases with W_RS/S_ < 1 did not differ significantly between unrelated and related strains (Fisher’s exact test, *P* = 0.27). Furthermore, costs were recessive (i.e., W_R/S_ < 1; W_RS/S_ = 1) in 8 cases (67%) and nonrecessive (i.e., W_R/S_ and W_RS/S_ < 1) in 4 cases (33%) for related strains (Table 2). Costs were recessive in 3 cases (25%) and nonrecessive in 9 cases (75%) for unrelated strains. There is a trend for the percentage of cases with nonrecessive costs to be higher for unrelated than related strains (Fisher’s exact test, *P* = 0.10).

#### 3.1.4. Hybrid Vigor and Higher Fitness in Resistant Than Susceptible Strains in Unrelated vs. Related Strains

We used the patterns of covariation between W_R/S_ and W_RS/S_ shown in Table 2 to evaluate the assumption that related strains are less affected than unrelated strains by inbreeding depression or differences between individuals used to generate the susceptible and resistant strains. Resistance to Bt is primarily conferred by loss-of-function mutations reducing the binding of toxins to midgut receptors [19,51,56]. Loss-of-function mutations mainly have deleterious effects [45], implying that resistance to Bt should rarely involve higher fitness in a resistant strain relative to susceptible strain in the absence of Bt. Accordingly, higher fitness in a resistant strain (and sometimes in both the resistant strain and F_1_ progeny) than in a susceptible strain could either indicate that the susceptible strain was more inbred than the resistant strain, or that field-collected individuals exposed to divergent selection regimes were used to generate the susceptible and resistant strains. On the other hand, a higher fitness in heterozygotes than in both parental strains is a hallmark of hybrid vigor resulting from inbreeding depression in both parental strains [45]. Thus, higher fitness in the F_1_ progeny than in both the susceptible and resistant strains indicates inbreeding depression in both parental strains. If related strains do have a common genetic background, higher fitness in the resistant than susceptible strain or hybrid vigor should be less common in related than unrelated strains. 

A surprising result is that the occurrence of hybrid vigor was not lower in cases with related strains (17%) than in cases with unrelated strains (13%) (Table 2). However, as expected, there was a trend for the occurrence of high fitness in the resistant strain relative to the susceptible strain to be lower with related strains (7%) relative to unrelated strains (24%) (Fisher’s exact test, *P* = 0.096). Thus, for cases of hybrid vigor but less so for cases of high fitness of the resistant strain relative to the susceptible strain, it appears that the genetic background of related strains was not properly homogenized when costs were measured. The number of generations elapsed between the measurement of costs and completion of the last backcross, or initiation of selection of the resistant strain from a susceptible strain, may sometimes have been large enough to allow divergence of the susceptible and resistant strain by genetic drift, causing inbreeding depression in both strains. Unfortunately, we could not test this hypothesis, because few studies reported the number of generations elapsed between the last backcross or onset of selection and the time cost experiments were conducted. 

We mentioned previously that costs in the resistant strains and F_1_ progeny were likely underestimated in *D. saccharalis*. The nine cases for this species from the US originated from six studies conducted in the same laboratory (Appendix A). W_R/S_ and W_RS/S_ were estimated for each case. These cases were based on unrelated strains according to our criterion, as the number of backcrosses used to homogenize the susceptible and resistant strains ranged from 0 to 3. Remarkably, 55% of cost estimates (n = 5) for *D. saccharalis* indicated higher fitness in both the resistant strain and F_1_ progeny than in the susceptible strain, which was significantly higher (Fisher’s exact test, *P* = 0.016) than the similar outcome for all studies (Table 2: 16%, 11 cases with higher fitness in resistant strain out of 68). Of the remaining W_R/S_ and W_RS/S_ estimates for *D. saccharalis*, 22% (n = 2) indicated hybrid vigor and 22% no costs. These patterns suggest that costs were underestimated in *D. saccharalis* due to inbreeding depression in the susceptible strains or divergent selection in individuals used to generate the susceptible and resistant strains (5 cases), or inbreeding depression of the susceptible and resistant strains (2 cases). 

### 3.2. Comparison of Incomplete Resistance between Cases with and without Practical Resistance

Survival of resistant strains on Bt plants relative to non-Bt plants (IR) was higher in cases with practical resistance than in cases without practical resistance. In the basic statistical model (see Appendix A for details and other non-significant effects tested), food type was marginally associated with IR (df = 1, 22, *F* = 3.91, *P* = 0.061). IR had a least squares mean value (back-transformed) of 0.45 (0.24–0.68) on cotton and 0.73 (0.55–0.91) on corn, indicating lower survival of resistant strains on Bt cotton relative to non-Bt cotton than on Bt corn relative to non-Bt corn. After controlling for the variation in food type, the species and country nested within food type were significantly associated with IR (Table 3) (df = 5, 22, *F* = 2.67, *P* = 0.050). Across diets, the average IR for cases without practical resistance (0.43, 95% CI = 0.11–0.74) was lower than for cases with practical resistance (0.76, 0.48–1.03) (one-tailed contrast, df = 1, 22, *F* = 5.45, *P* = 0.014).

### 3.3. Modeling the Effects of Costs and Incomplete Resistance on Evolution of Practical Resistance

In simulations, resistance evolved much slower with recessive inheritance of resistance (*h* = 0) than nonrecessive resistance (*h* = 0.26) (Figure 1 and Table 4). For example, with a 10% refuge, no cost, and IR = 1, resistance evolved in 129 generations with *h* = 0 and 7 generations with *h* = 0.26, an 18-fold difference (Table 4). Also with a 10% refuge, resistance evolved in 321 generations using the parameter values for cases without practical resistance (*h* = 0, cost = 30%, IR = 0.43) versus 8 generations using the parameter values for cases with practical resistance (*h* = 0.26, cost = 14%, IR = 0.76), which is a 40-fold difference. Thus, in addition to the effect of the dominance of resistance *(h* = 0 versus 0.26), the differences in cost and IR more than doubled the time for resistance to evolve.

Results of simulations varying IR and cost separately show that with other parameters fixed, the time for resistance to evolve was affected more by the observed difference in IR between cases with and without practical resistance (0.76 and 0.43, respectively) than by the observed difference in costs for cases with and without practical resistance (14 and 30%, respectively; Figure 1 and Table 4). For example, with a 5% refuge, *h* = 0, and IR = 0.76, resistance evolved in 91 or 92 generations with cost = 14% or 30%, respectively. However, with a refuge of 5%, *h* = 0, and IR = 0.43, resistance evolved in 159 or 162 generations with cost = 14% or 30%, respectively (Figure 1A, Table 4). Thus, the increase in the simulated time to evolve resistance was over 74% from the difference in IR but less than 2% from the difference in cost. In general, increasing only cost across the range of values examined here had little effect on the time for resistance to evolve, with exceptions when *h* was 0.26 and the refuge percentage was 60 or 70% (Figure 1B, Table 4). Overall, the simulation results suggest that in the evolution of practical resistance to single-toxin Bt crops, variation in the dominance of resistance and IR played a greater role than variation in the costs.

## 4. Conclusion and Future Prospects

Our results identify a resistance syndrome characterizing insect cases with and without evolution of practical resistance. This syndrome involves a higher dominance of resistance, lower fitness costs, and higher survival of resistant insects on Bt crops relative to non-Bt crops in cases with practical resistance than without practical resistance. This syndrome likely has implications for evolution of resistance to single-toxin and pyramided crops because it involves key traits underlying success of the refuge strategy for both types of crops [19,20,29]. 

Inherent susceptibility to Bt toxins could underlie this covariation among traits. The dominance of resistance generally decreases at high concentrations of Bt toxins [18,43]. Thus, for a given toxin concentration, pests with low inherent susceptibility exhibit more dominant resistance than pests with high inherent susceptibility [20,24,25]. Furthermore, mutations that cause small decreases in susceptibility could be sufficient to overcome Bt toxins in pests with low inherent susceptibility, e.g., [57]. Such mutations may also cause smaller fitness costs than mutations that confer greater decreases in susceptibility [28,58]. Individuals with low inherent susceptibility to Bt toxins might also be more likely to have similar fitness on Bt and non-Bt crops. 

Our simulations indicate that the propensity to evolve practical resistance to single-toxin crops is better explained by differences in the dominance of resistance and incomplete resistance than differences in costs. We also found evidence that IR was higher on Bt corn than Bt cotton, indicating that survival on the Bt crop relative to its non-Bt counterpart was higher for corn than cotton. This is consistent with the greater percentage of cases with practical resistance for Bt corn (19/38 = 50%) than Bt cotton (6/31 = 19%, Fisher’s exact test, *P* = 0.012) [14]. Previous simulations indicate that costs are more effective for delaying the evolution of resistance to pyramided than single-toxin crops, because individuals resistant to a single toxin do not gain a fitness advantage on pyramids, due to redundant killing, but are nevertheless negatively affected by recessive fitness costs in refuges [32,59]. Accordingly, differences in costs between cases with and without practical resistance could have a greater influence on the evolution of resistance to pyramided than single-toxin crops. As more data become available for pest species monitored for resistance to Bt crops, it will be useful to determine if larger costs and lower survival on Bt crops relative to non-Bt crops are associated with resistance to multiple-toxin crops compared to single-toxin crops. We expect that the resistance syndrome for pyramided crops will include differences in redundant killing, because this trait depends on the effectiveness of Bt toxins [19,47].

We found that studies using unrelated susceptible and resistant strains tended to overestimate the dominance of costs and the frequency of cases with higher fitness in resistant strains and F_1_ progeny relative to susceptible strains. However, use of related strains did not reduce the occurrence of hybrid vigor. This indicates that the precision of fitness component studies will be improved by using related strains and minimizing the number of generations elapsed between the onset of cost experiments and initiation of the selection of resistant strains from susceptible strains or completion of backcrosses. Minimizing delays between backcrosses would also be important. Reporting the timing of these events is needed to better evaluate potential effects of inbreeding depression on cost estimates. 

We used the statistical significance of at least one comparison of fitness components between a resistant strain or the F_1_ progeny and a susceptible strain to determine that costs were present. If fitness components did not differ significantly between a resistant strain or the F_1_ progeny and a susceptible strain due to low statistical power rather than an absence of costs per se, this could have underestimated the prevalence of costs. We also selected fitness components that differed significantly between a resistant strain or the F_1_ progeny and a susceptible strain to estimate W_R/S_ or W_RS/S_. This could have underestimated costs or fitness advantages in resistant strains or the F_1_ progeny if the lack of statistical significance was due to low statistical power. Sample size is not expected to differ consistently between studies of cases with and without practical resistance. Accordingly, use of statistical significance as a criterion to determine the presence of costs or select fitness components to estimate costs should not have affected the difference in costs between cases with and without practical resistance reported here. 

In contrast to previous results [28], we did not find greater costs on host plants than artificial diet, or a positive association between resistance levels and costs (Appendix A). These differences between studies may have occurred because we investigated different sets of insect species, types of resistance (Gassmann et al. [28] considered resistance to single Bt toxins and Bt sprays together), or both. Gassmann et al. [28] recommended that modelers use a recessive fitness cost of 25% as a standard value when relevant data are lacking. Here, we propose using a larger recessive cost for cases without practical resistance (30%, range based on 95% confidence interval = 19–42%) and a smaller recessive cost for cases with practical resistance (14%, range 4–24%). Additionally, we recommend using values of incomplete resistance of 0.76 (range 0.48–1) and 0.43 (range 0.11–0.74) for cases with and without practical resistance in simulations, respectively. For species not currently monitored for resistance, the average recessive cost (20%, range 12–28%) and incomplete resistance (0.62, range 0.37–0.86) for species with and without practical resistance could be used in simulations. Although a previous study [26] did not find significant costs associated with resistance to Bt in pests with and without practical resistance, several pitfalls may have affected this finding. These include small sample size, failure to account for factors not related to resistance, and use of cost estimates based on r_m_ that underestimate costs. 

Knowledge of the history of resistance evolution in particular pest species is useful to design effective resistance management programs [60]. As more information becomes available on pests that evolve practical resistance [14] and the syndrome associated with this resistance, it will become increasingly possible to consider characteristics of pest species targeted by Bt crops to proactively improve resistance management strategies. 

## Figures and Tables

**Figure 1 insects-14-00214-f001:**
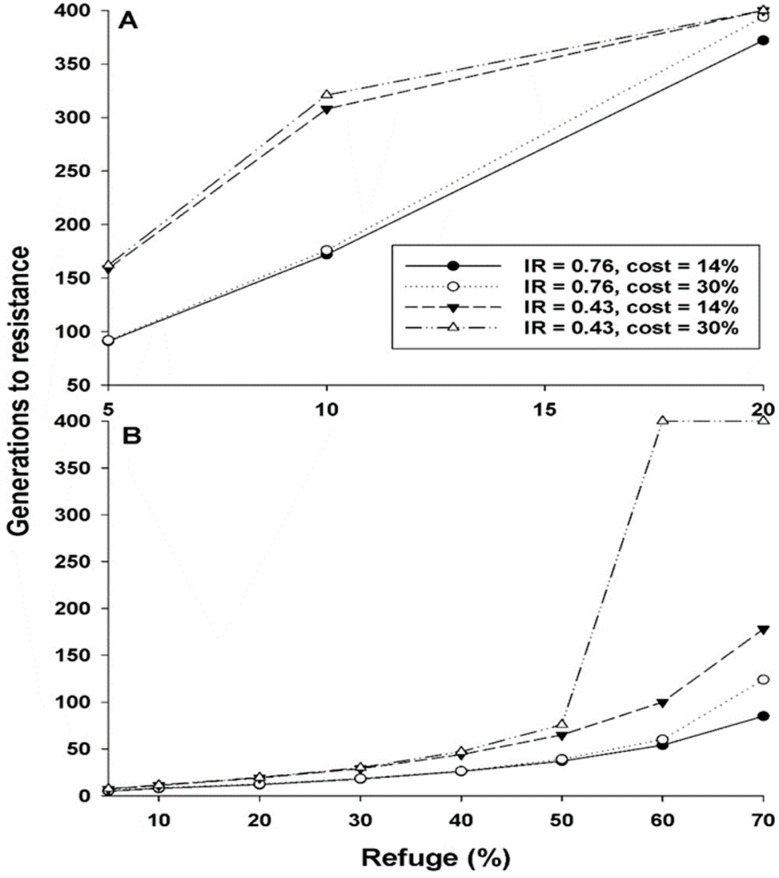
Effects of costs and incomplete resistance (IR) on evolution of resistance in simulations with (**A**) Recessive resistance (*h* = 0) and (**B**) Nonrecessive resistance (*h* = 0.26). Values for IR and costs correspond to observed means from cases with practical resistance (IR = 0.76, cost = 14%) and without practical resistance (IR = 0.43, cost = 30%). Generations to resistance are the generations for the resistance allele frequency to increase from the initial value of 0.001 to 0.50. Simulation runs were stopped if resistance allele frequency did not reach 0.50 after 400 generations.

**Table 1 insects-14-00214-t001:** Fitness ratios (W_R/S_ and W_RS/S_) and associated fitness costs of resistance.

Species	Country ^a^	Food	W_R/S_ ^b^	Cost for Resistant Strain (%)	W_RS/S_ ^b^	Cost for F_1_ Progeny (%)	Practical Resistance ^c^
*C. virescens*	US	Diet	0.74 (0.25–1.37)	26	--	--	No
*D. saccharalis*	US	Diet	1.09 (0.78–1.44)	0	1.08 (0.95–1.24)	0	No
*H. armigera*	CH	Diet	0.53 (0.35–0.73)	47	--	--	No
*H. zea*	US	Diet	1.08 (0.73–1.46)	0	1.09 (0.93–1.28)	0	Yes
*O. nubilalis*	US	Diet	0.64 (0.36–0.98)	36	1.12 (0.99–1.29)	0	No
*P. gossypiella*	IN	Diet	0.78 (0.41–1.23)	22	0.92 (0.79–1.08)	8	Yes
*P. gossypiella*	US	Diet	0.79 (0.58–1.00)	21	0.94 (0.86–1.03)	6	No
*S. frugiperda*	BR	Diet	0.42 (0.00–0.96)	58	--	--	Yes
*S. frugiperda*	US	Diet	0.89 (0.67–1.13)	11	1.03 (0.94–1.12)	0	Yes
*B. fusca*	SA	Corn	0.93 (0.53–1.39)	7	--	--	Yes
*D. saccharalis*	US	Corn	1.14 (0.84–1.45)	0	1.10 (0.99–1.23)	0	No
*D. v. virgifera*	US	Corn	0.85 (0.71–0.99)	15	0.99 (0.85–1.16)	1	Yes
*O. nubilalis*	US	Corn	0.76 (0.26–1.38)	24	0.96 (0.77–1.20)	4	No
*S. frugiperda*	BR	Corn	0.94 (0.73–1.14)	6	1.08 (1.00–1.16)	0	Yes
*S. frugiperda*	US	Corn	0.92 (0.69–1.16)	8	1.06 (0.95–1.18)	0	Yes
*H. armigera*	AU	Cotton	0.64 (0.36–0.96)	36	0.88 (0.75–1.03)	12	No
*H. zea*	US	Cotton	0.92 (0.66–1.20)	8	0.97 (0.86–1.09)	3	Yes
*P. gossypiella*	US	Cotton	0.80 (0.60–1.00)	20	0.90 (0.83–0.97)	10	No
*S. frugiperda*	BR	Cotton	0.74 (0.38–1.17)	26	1.00 (0.86–1.17)	0	Yes
*S. frugiperda*	US	Cotton	0.99 (0.58–1.46)	1	1.10 (0.93–1.29)	0	Yes

^a^. Australia: AU, United States: US, China: CH, Brazil: BR, India: IN, South Africa: SA. ^b^. Back-transformed least squares means and 95% confidence intervals for the fitness ratio relative to a susceptible strain for a resistant strain (W_R/S_) and F_1_ progeny from mating between a resistant and susceptible strain (W_RS/S_). ^c^. Yes means practical resistance to at least one toxin [14].

**Table 2 insects-14-00214-t002:** Outcome of fitness comparisons between the resistant strain and F_1_ progeny relative to the susceptible strain for cases involving related strains (n = 30) or unrelated strains (n = 38). The percentage of cases (number in parentheses) for each outcome is shown. Fitness comparisons are based on the values of W_R/S_ and W_RS/S_ and do not necessarily reflect statistically significant differences.

Outcome	Strain Relatedness	
	Related	Not Related
Recessive cost ^a^	27 (8)	8 (3)
Nonrecessive cost ^b^	13 (4)	24 (9)
No cost ^c^	33 (10)	26 (10)
Hybrid vigor ^d^	17 (5)	13 (5)
High fitness in resistant strain ^e^	7 (2)	24 (9)
Fitness: F_1_ < susceptible and resistant > susceptible ^f^	3 (1)	5 (2)

^a^. W_R/S_ < 1 and W_RS/S_ = 1. ^b^. W_R/S_ and W_RS/S_ < 1. ^c^. W_R/S_ and W_RS/S_ = 1. ^d^. W_RS/S_ > 1 and W_R/S_ < W_RS/S_. ^e^. W_R/S_ > 1 and W_RS/S_ = 1 or W_R/S_ and W_RS/S_ > 1 but W_R/S_
*≥* W_RS/S_. ^f^. W_RS/S_ < 1 and W_R/S_ > 1.

**Table 3 insects-14-00214-t003:** Incomplete resistance (IR) for species from different countries fed Bt and non-Bt corn or Bt and non-Bt cotton.

Species	Country ^a^	Food	IR ^b^	Practical Resistance ^c^
*B. fusca*	SA	Corn	0.93 (0.38–1.60)	Yes
*D. saccharalis*	US	Corn	0.34 (0.10–0.64)	No
*D. v. virgifera*	US	Corn	0.69 (0.54–0.86)	Yes
*S. frugiperda*	BR	Corn	1.00 (0.76–1.27)	Yes
*H. armigera*	AU	Cotton	0.48 (0.25–0.75)	No
*H. zea*	US	Cotton	0.41 (0.10–0.79)	Yes
*P. gossypiella*	US	Cotton	0.46 (0.03–1.02)	No

^a^. Australia: AU, United States: US, Brazil: BR, South Africa: SA. ^b^. Back-transformed least squares means and 95% confidence intervals for IR. ^c^. Yes means practical resistance to at least one toxin [14].

**Table 4 insects-14-00214-t004:** Simulated effects of cost and incomplete resistance (IR) on evolution of resistance. Inheritance of resistance was recessive (*h* = 0) or nonrecessive (*h* = 0.26). For each value of a recessive cost (0, 14, or 30%), IR was 1, 0.76, or 0.43, respectively. For each combination of parameter values, we report the number of generations for the resistance allele frequency to increase from its initial value of 0.001 to more than 0.50. Results in bold correspond to cases of pests without practical resistance (*h* = 0, cost = 30%, IR = 0.43) and with practical resistance (*h* = 0.26, cost = 14 %, IR = 0.76).

Refuge%	No Cost			Cost = 14%			Cost = 30%		
IR =	1	0.76	0.43	1	0.76	0.43	1	0.76	0.43
*h* = 0									
5	70	90	156	70	91	159	71	92	**162**
10	129	169	296	131	172	308	133	176	**321**
20	270	355	>400	280	372	>400	292	394	**>400**
*h* = 0.26									
5	5	5	7	5	**5**	7	5	5	7
10	7	8	11	7	**8**	11	7	8	11
20	10	12	19	10	**12**	19	10	12	19
30	15	18	29	15	**18**	29	15	18	30
40	21	25	42	21	**26**	44	21	26	47
50	29	35	60	30	**37**	65	31	39	76
60	41	50	88	43	**54**	100	46	60	>400
70	61	76	133	66	**85**	178	77	124	>400

## Data Availability

Data used for this study are available in the manuscript or in Appendix A and Appendix A posted online at https://www.mdpi.com/article/10.3390/insects14030214/s1.

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
