# Peer review of "Fitness Costs and Incomplete Resistance Associated with Delayed Evolution of Practical Resistance to Bt Crops"

_insects, 2023, doi:10.3390/insects14030214_

Round 1
Reviewer 1 Report
The manuscript “Fitness costs and incomplete resistance associated with delayed evolution of practical resistance to Bt crops” summarized a great deal of literatures to demonstrate the classical hypothesis that fitness costs and incomplete resistance are two important factors affecting evolution of practical resistance to Bt crops by target pests. The authors conclude that both lower fitness costs and more complete resistance contribute to evolution of practical resistance and the latter plays a greater role than the former. The statistical evidence is compelling and the manuscript is well written. The paper is ready for publication after correcting some typos in the proof stage (such as Line 329, "3.4" should be "3.1.4").
Author Response
We thank this reviewer for the positive comments.
We have corrected the typo on line 329 as requested.
Reviewer 2 Report
The review collected and analyzed data from the literature to evaluate the association between practical resistance to Bt crops and two pest traits: fitness costs and 24 incomplete resistance.
This review can also help relevant researchers systematically understand the relevant research progress and the mechanism of insect resistance. The review data is complete and clear, and the statistical analysis is reasonable.
Comments
In order to make broader researchers (Researchers not in this field) understand relevant research progress, it is suggested to define key concepts, such as Fitness costs; Incomplete resistance; Delayed evolution;Practical resistance
Author Response
We thank this reviewer for the positive comments.
As requested we have made definitions of fitness costs, incomplete resistance, and redundant killing more explicit. We have also added a definition for inbreeding depression.
Reviewer 3 Report
The paper is well written. The authors clearly state the steps implemented to compile the studies used in the analysis regarding practical resistance to Bt crops. I think it is a valuable compilation to the scientific literature.
Author Response
We thank this reviewer for the positive comments.